# Note on Limit-Periodic Solutions of the Difference Equation $x_{t+1} - [h(x_t) + \lambda]x_t = r_t, \lambda > 1$

**Jan Andres** [1,*]  **and Denis Pennequin** [2]

[1]   Department of Mathematical Analysis and Applications of Mathematics, Faculty of Science,
     Palacký University, 17. listopadu 12, 771 46 Olomouc, Czech Republic
[2]   Centre PMF, Laboratory SAMM, Université Paris I Panthéon—Sorbonne, 90, Rue de Tolbiac,
     75 634 Paris CEDEX 13, France; pennequi@univ-paris1.fr
*    Correspondence: jan.andres@upol.cz; Tel.: +420-585-634-602

**Abstract:** As a nontrivial application of the abstract theorem developed in our recent paper titled "Limit-periodic solutions of difference and differential systems without global Lipschitzianity restricitions", the existence of limit-periodic solutions of the difference equation from the title is proved, both in the scalar as well as vector cases. The nonlinearity $h$ is not necessarily globally Lipschitzian. Several simple illustrative examples are supplied.

**Keywords:** limit-periodic solutions; difference equations; exponential dichotomy; strong nonlinearities; effective existence criteria; population dynamics

**MSC:** 39A11; 39A24; 42A75

## 1. Introduction

As far as we know, in spite of the intensive studies of limit-periodic (especially Schrödinger-type) operators (see, e.g., [1,2], and the references therein), the results about the existence of limit-periodic solutions to nonlinear differential and difference equations are very rare (see e.g., [3–8]). For some related periodicity and almost-periodicity problems and applications, see, e.g., [9–12], and the references therein. As already pointed out in [5], since the space of limit-periodic sequences is (unlike for limit-periodic functions) Banach (cf. [4,13]), the existence criteria for limit-periodic solutions of difference equations are significantly simpler than those for differential equations. Nevertheless, an obstruction related to the absence of global Lipschitzianity restrictions, imposed on the given right-hand sides of equations under consideration, remains also in the discrete case. For the recently investigated continuous case, see [6] and the references therein.

Hence, the aim of the present note is to obtain, by means of our technique (see [5], Theorem 3.2, resp. Corollary 3.3), which we state below in the form of Proposition 2, the effective solvability criteria of the equation from the title. Its scalar and vector cases will be treated separately. Let us note that, in particular cases, the equation from the title can describe discrete population models for a single species. For instance, if $h(x) := -\mu x + \mu - \lambda, \mu > 0$, then we get the *forced logistic equation*. For more details, see e.g., ([14], Chapter 1, [15], Chapter 2).

## 2. Preliminaries and Auxiliary Results

At first, we will recall the notion of a limit-periodic sequence and its basic properties.

**Definition 1.** *A sequence $\underline{x} := \{x_t\} \in (\mathbb{R}^n)^{\mathbb{Z}}$, where $\mathbb{R}$ and $\mathbb{Z}$ denote respectively the sets of reals and integers, is called limit-periodic if there exists a family of periodic sequences $\underline{x}^k := \{x_t^k\}$, $k \in \mathbb{N}$ ($\mathbb{N}$ denotes the set of positive integers), such that $\lim_{k \to \infty} x_t^k = x_t$, uniformly w.r.t. $t \in \mathbb{Z}$.*

It is well known (see e.g., [5]) that Definition 1 is equivalent to the following definition of a semi-periodic sequence (cf. [13]).

**Definition 2.** *A sequence $\underline{x} := \{x_t\} \in (\mathbb{R}^n)^{\mathbb{Z}}$ is called semi-periodic if*

$$\forall \varepsilon > 0, \ \exists T \in \mathbb{N}, \ \forall m \in \mathbb{Z}, \ \forall k \in \mathbb{Z}, \quad |x_{k+mT} - x_k| \leq \varepsilon.$$

**Remark 1.** *Since the uniform and Stepanov norms, namely $\| \cdot \|_\infty$ and $\| \cdot \|_{S_T^p}$, where*

$$\|\underline{x}\|_\infty := \sup_{t \in \mathbb{Z}} |x_t|,$$

$$\|\underline{x}\|_{S_T^p} := \sup_{m \in \mathbb{Z}} \left( \frac{1}{T+1} \sum_{t=m}^{m+T} |x_t|^p \right)^{\frac{1}{p}}, \ T \in \{0\} \cup \mathbb{N}, \ p \geq 1,$$

*were shown in ([16], Proposition 1 and Remark 2) to be equivalent, both Definitions 1 and 2 can be easily reformulated in terms of Stepanov limit-periodic and Stepanov semi-periodic sequences by means of the Stepanov norm $\| \cdot \|_{S_T^p}$, $p \geq 1$.*

Summing up, it will be convenient to recall the following proposition (cf. [5], Proposition 2.12).

**Proposition 1.** *The following properties of a sequence $\{x_t\} \in (\mathbb{R}^n)^{\mathbb{Z}}$ are equivalent:*

(i) *$\{x_t\}$ is uniformly limit-periodic,*
(ii) *$\{x_t\}$ is (Stepanov) $S_T^p$-limit-periodic,*
(iii) *$\{x_t\}$ is uniformly semi-periodic,*
(iv) *$\{x_t\}$ is (Stepanov) $S_T^p$-semi-periodic.*

*Moreover, the set of all (Stepanov) limit-periodic, resp. (Stepanov) semi-periodic sequences $\{x_t\} \in (\mathbb{R}^n)^{\mathbb{Z}}$, endowed with the uniform or Stepanov norms $\| \cdot \|_\infty$ or $\| \cdot \|_{S_T^p}$, $p \geq 1$, is a Banach space.*

Now, let us proceed to the difference system

$$x_{t+1} = f(x_t) + p_t, \tag{1}$$

where $f \in C^1(\mathbb{R}^n, \mathbb{R}^n)$ and $\{p_t\} \in (\mathbb{R}^n)^{\mathbb{Z}}$ is a (Stepanov) limit-periodic sequence. Let us also consider the associated one-parameter family of systems

$$x_{t+1} = f(x_t) + p_t^N, \tag{2}$$

where the class of $T_k$-periodic ($T_k > 0$) sequences $\{p_t^N\} \in (\mathbb{R}^n)^{\mathbb{Z}}$, $N \in \mathbb{N}$, converges uniformly to $\{p_t\}$.

Our technique in ([5], Corollary 3.7) will be stated here in the form of the following proposition.

**Proposition 2.** *Assume still that*

(i) *for each fixed N, system (2) admits a $T_k$-periodic solution $\{x_t^N\}$,*
(ii) *$\sup_{N \in \mathbb{N}} \|\{x_t^N\}\|_\infty < \infty$,*
(iii) *if $A_t^N$ is the Jacobian matrix of $f$ at $x_t^N$, then there exists a non-singular solution of the homogeneous system*

$$y_{t+1} = A_t^N y_t, \tag{3}$$

*which satisfies the exponential dichotomy, for all sufficiently large values of N, with common constants K and α, characterizing the exponential dichotomy.*

   *Then, system* (1) *possesses a uniformly limit-periodic solution.*

   Let us recall the definition of an exponential dichotomy for Equation (3). Introducing the *resolvent* $R\colon \mathbb{Z}^2 \to \mathcal{M}_n$, where $\mathcal{M}_n$ denotes the space of real $n \times n$ matrices, and

$$R(t,s) := \begin{cases} A_{t-1}\ldots A_s, & \text{for } t > s, \\ A_s^{-1}\ldots A_{t-1}^{-1}, & \text{for } t < s, \\ I_n, & \text{for } t = s, \end{cases}$$

where $I_n$ is the identity matrix, it has the semi-group property. Taking $\Phi_t := R(t,0)$, we say that Equation (3) satisfies an *exponential dichotomy* if there exists a projection $P$ ($P^2 = P$) and constants $K > 0, \alpha \in (0,1)$ such that

$$\begin{cases} |\Phi_t P \Phi_s^{-1}| \leq K\alpha^{t-s}, & \text{for } t \geq s, \\ |\Phi_t (I_n - P)\Phi_s^{-1}| \leq K\alpha^{-(t-s)}, & \text{for } t \leq s. \end{cases} \tag{4}$$

   Let us note that $\Phi_t P \Phi_s^{-1} y$ is the $t$-iterated image of the projection by $P$ of the solution of Equation (3) such that $y_s = y$. In the stable case, $P = I_n$, and so $\Phi_t P \Phi_s^{-1} = R(t,s)$, when $t \geq s$, and $\Phi_t (I_n - P)\Phi_s^{-1} = 0$, when $t \leq s$.

**Remark 2.** *For some alternative definitions of an exponential dichotomy for Equation* (3), *see, e.g.,* [17–19]. *In particular, Palmer gives in* [18] *a finite-time condition for an exponential dichotomy. In fact, all these conditions were formulated for a more general class of (uniformly) almost-periodic homogeneous systems* (3).

   On this basis, we can define the associated *Green function* $G\colon \mathbb{Z}^2 \to \mathcal{M}_n$, where (see, e.g., [5], and the references therein)

$$G(t,s) := \Phi_t \left( P l_{t\geq s} + (I_n - P) l_{t\leq s} \right) \Phi_s^{-1} = \begin{cases} \Phi_t P \Phi_s^{-1}, & \text{for } t > s, \\ I_n, & \text{for } t = s, \\ \Phi_t (I_n - P)\Phi_s^{-1}, & \text{for } t < s, \end{cases} \tag{5}$$

and

$$l_{t\geq s} := \begin{cases} 1, & \text{for } t \geq s, \\ 0, & \text{for } t < s. \end{cases}$$

## 3. Limit-Periodic Solutions: Scalar Case

   Consider the equation from the title in the scalar case ($n = 1$), i.e.,

$$x_{t+1} - [h(x_t) + \lambda] x_t = r_t, \tag{6}$$

where $\lambda > 1, h \in C^1(\mathbb{R}, \mathbb{R})$ and $\underline{r} = \{r_t\} \colon \mathbb{Z} \to \mathbb{R}$ is a (Stepanov or) uniformly limit-periodic sequence (see Definition 1 and Proposition 1).

   At first, let us deal with the case, when $\underline{r}$ is periodic. Since we would like to obtain a periodic solution for Equation (6), we associate to it its Schauder-like parametrization, namely

$$x_{t+1} - [h(q_t) + \lambda] x_t = r_t, \tag{7}$$

where $\underline{q} \in Q_D := \{\underline{p} \in \mathbb{R}^{\mathbb{Z}}, \|\underline{p}\|_\infty \leq D\}$.

Consider still the homogeneous equation, obtained by the linearization of Equation (6) at $\{q_t\}$, i.e.,

$$x_{t+1} - \left[ h'(q_t)q_t + h(q_t) + \lambda \right] x_t = 0. \tag{8}$$

Let us assume that there exists a constant $D > 0$ such that

$$\forall x \in [-D, D] \colon h(x) \geq 0 \text{ and } h'(x)x + h(x) \geq 0, \tag{9}$$

jointly with

$$\|\underline{r}\|_\infty \leq \frac{\lambda - 1}{\lambda + 1} D. \tag{10}$$

We are ready to formulate the first main theorem (for the scalar case), when applying Proposition 2.

**Theorem 1.** *Let $\lambda > 1$ and let there exist a constant $D > 0$ such that condition (9) holds for $h \in C^1(\mathbb{R}, \mathbb{R})$, jointly with condition (10) for a (Stepanov or) uniformly limit-periodic sequence $\{r_t\} \colon \mathbb{Z} \to \mathbb{R}$. Then, Equation (6) admits a uniformly limit-periodic solution $\underline{z}$ satisfying*

$$\|\underline{z}\|_\infty \leq \frac{\lambda + 1}{\lambda - 1} \|\underline{r}\|_\infty. \tag{11}$$

**Proof.** Observe that, under the assumption (9), the homogeneous Equation (8) exhibits an exponential dichotomy with constants $K = 1$ and $\alpha = 1/\lambda$ because

$$|h'(q_t)q_t + h(q_t) + \lambda| = h'(q_t)q_t + h(q_t) + \lambda \geq \lambda,$$

by which

$$|x_t| = \frac{1}{|h'(q_t)q_t + h(q_t) + \lambda|} |x_{t+1}| \leq \frac{1}{\lambda} |x_{t+1}|.$$

Moreover, Equation (7) admits a unique entirely bounded solution $\underline{u}$ which takes the form

$$u_t = \sum_{\ell \in \mathbb{Z}} G_q(t, \ell) r_{\ell-1},$$

where $G_q$ is the Green function for Equation (7), where $r_t = 0$, (see Formula (5)). By the standard calculations, we obtain that (see e.g., [5])

$$\|\underline{u}\|_\infty \leq K \frac{1 + \alpha}{1 - \alpha} \|\underline{r}\|_\infty = \frac{\lambda + 1}{\lambda - 1} \|\underline{r}\|_\infty.$$

If $\underline{r}$ is $T_k$-periodic, then so must be $\underline{u}$ (see e.g., [20], Theorem 2.6).

For each $k \in \mathbb{N}$, we introduce

$$Q_k := \left\{ \underline{p} \in \mathbb{R}^{\mathbb{Z}}, \ \underline{p} \text{ is } T_k\text{-periodic and } \left\| \underline{p} \right\|_\infty \leq D \right\},$$

and the operator $\mathcal{T}_k \colon Q_k \to \mathbb{R}^{\mathbb{Z}}$, where

$$\mathcal{T}_k(\underline{q}) := \sum_{\ell \in \mathbb{Z}} G_q(t, \ell) r_{\ell-1}.$$

One can easily check that this operator is continuous and compact ($Q_k \cap \mathbb{R}^{\mathbb{Z} \cap [0, T_k]}$ is compact), and such that $\mathcal{T}_k(Q_k) \subset Q_k$, provided condition (10) holds.

Thus, for a given $\underline{r}$, we take $D$ such that $D \geq \frac{\lambda+1}{\lambda-1} \|\underline{r}\|_\infty$, and the set $Q_k$ as above. Applying the well known Brouwer fixed point theorem, $\mathcal{T}_k$ possesses a fixed point $\varphi_k \in Q_k$, which represents a $T_k$-periodic solution of Equation (6), where $\underline{r}$ is $T_k$-periodic. Moreover, $\sup_{k \in \mathbb{N}} \|\varphi_k\|_\infty \leq D$.

Now, let us proceed to a limit-periodic sequence $\underline{r}$. According to Definition 1, it is a limit of a family of periodic sequences $\{\underline{r_k}\}$. We take $D > \frac{\lambda+1}{\lambda-1} \sup_{k \in \mathbb{N}} \|\underline{r_k}\|_\infty$. Since we have the exponential dichotomy with the same constants for each perturbated system, all the assumptions of Proposition 2 are satisfied. Thus, we obtain the existence of a limit-periodic solution $\underline{z}$ of Equation (6) satisfying the inequality (11), which completes the proof. □

**Remark 3.** *Taking $H(x) := h'(x)x + h(x)$, we can see that if h is even (resp. odd), then H must be also. Thus, if h is odd, then the only situation in order function H could satisfy the assumption (9) occurs, provided $h(x) = 0$, for each $x \in [0, D]$, which does not have much meaning. On the other hand, if h is even, then H must be too, and it is sufficient to satisfy the inequality in condition (9) on $(0, D]$. One can easily check that the function h satisfies this assumption on $(0, D]$ if and only if the function $x \mapsto xh(x)$ ($\geq 0$) is non-decreasing. We can see then that the assumption (9) is in fact a local non-decreasing property.*

We can give some illustrative examples in order condition (9) to be satisfied with an implicit or explicit $D$.

**Example 1** (with an implicit $D$)**.** *Assume that there is an expansion of h arround 0,*

$$h(x) := a_n x^n + o(x^n), \ a_n > 0,$$

*where n is a (strictly) positive even number. In such a case, we have $h'(x)x + h(x) = x^n (na_n + o(1))$, with $na_n > 0$. Unfortunately, an implicit character of D does not lead to an effective solvability criterium here.*

**Example 2** ($D$ can be any positive real number)**.** *Let h be a polynomial of an even degree, namely*

$$h(x) := \sum_{k=0}^{N} a_k x^{2k},$$

*where $a_k \geq 0$, for every k.*

*More generally, we can consider the case of the following even Taylor expansion:*

$$h(x) := \sum_{k=0}^{+\infty} a_k x^{2k},$$

*where $a_k \geq 0$, for every k. Then, any D (strictly) smaller than the radius of convergence of the series exists. For instance, we can consider the function*

$$h(x) := \frac{1}{1 - x^2},$$

*with $D \in (0, 1)$. In this case, condition (10) can be reduced to*

$$\|\underline{r}\|_\infty < \frac{\lambda - 1}{\lambda + 1},$$

*provided $D = 1 - \varepsilon$ holds with a sufficiently small $\varepsilon > 0$.*

## 4. Limit-Periodic Solutions: Vector Case

In this section, we will consider again Equation (6), but this time in a vector case.

Hence, let $\lambda > 1$, $h \in \mathrm{C}^1(\mathbb{R}^n, \mathbb{R})$ and $\underline{r} = \{r_t\} \colon \mathbb{Z} \to \mathbb{R}^n$ be a (Stepanov or) uniformly limit-periodic sequence. As before, we associate to this equation its Schauder-like parametrization (7), and the homogeneous equation

$$x_{t+1} - [(\nabla h(q_t) \cdot x_t)q_t + h(q_t)x_t + \lambda x_t] = 0, \tag{12}$$

where this time $q \in Q_D := \{p \in (\mathbb{R}^n)^{\mathbb{Z}}, \|p\|_\infty \leq D\}$. We also introduce, for each $q \in \mathbb{R}^n$, the (continuous) linear mapping $L_q \colon \mathbb{R}^n \to \mathbb{R}^n$, where

$$L_q \colon x \mapsto (\nabla h(q) \cdot x)q + h(q)x. \tag{13}$$

Letting $B_D := \{x \in \mathbb{R}^n, \|x\| \leq D\}$, we will assume that one of the following assumptions holds. Let us note that the first one is a vector analogy of the scalar case. Moreover, it is equivalent to impose this assumption either for all $x \in \mathbb{R}^n$, or on the ball $B_D$:

$$\exists D > 0, \forall x \in \mathbb{R}^n, \forall q \in B_D, \quad h(q) \geq 0 \text{ and } L_q(x) \cdot x \geq 0, \tag{14}$$

$$\exists \beta > 1, \exists D > 0, \forall x \in \mathbb{R}^n, \forall q \in B_D, \quad h(q) \geq 0 \text{ and } \|L_q(x) + \lambda x\| \geq \beta \|x\|. \tag{15}$$

Assume still that

$$\|r\|_\infty \leq \frac{\lambda - 1}{\lambda + 1} D, \text{ or (in the case of condition (15)) } \|r\|_\infty \leq \frac{\beta - 1}{\beta + 1} D. \tag{16}$$

We are ready to formulate the second main theorem (for the vector case).

**Theorem 2.** *Let $\lambda > 1$ and let there exist a constant $D > 0$ such that either condition (14) or condition (15), with a suitable constant $\beta > 1$, hold for $L_q$ defined in the mapping (13), where $h \in C^1(\mathbb{R}^n, \mathbb{R})$. Let condition (16) still hold for a (Stepanov or) uniformly limit-periodic sequence $\{r_t\} \colon \mathbb{Z} \to \mathbb{R}^n$. Then, Equation (6) admits a uniformly limit-periodic solution $\underline{z}$ satisfying*

$$\|\underline{z}\|_\infty \leq \frac{\lambda + 1}{\lambda - 1} \|r\|_\infty, \text{ resp. } \|\underline{z}\|_\infty \leq \frac{\beta + 1}{\beta - 1} \|r\|_\infty.$$

**Proof.** Under the assumptions (14) or (15), the homogeneous Equation (12) exhibits an exponential dichotomy. Indeed, in the first case, for any solution, we have

$$\|x_{t+1}\| \cdot \|x_t\| \geq x_{t+1} \cdot x_t \geq L_q(x_t) \cdot x_t + \lambda x_t \cdot x_t \geq \lambda \|x_t\|^2.$$

Thus, we receive the exponential dichotomy with constants $K = 1$ and $\alpha = 1/\lambda$. In the second case, a simple calculation leads to the exponential dichotomy with constants $K = 1$ and $\alpha = 1/\beta$. To consider both situations together, let us replace $\lambda$ by $\beta$ in the first case. The unique entirely bounded solution $\underline{u}$ of Equation (7) satisfies this time the inequality

$$\|\underline{u}\|_\infty \leq \frac{\beta + 1}{\beta - 1} \|r\|_\infty.$$

If $\underline{r}$ is $T_k$-periodic, then so must be $\underline{u}$ (see again [20], Theorem 2.6).

Proceeding in a quite analogous way as in the scalar case in Section 3, we can prove the existence of a $T_k$-periodic solution $\varphi_k$ of Equation (6), where $\underline{r}$ is $T_k$-periodic and such that $\sup_{k \in \mathbb{N}} \|\varphi_k\|_\infty \leq D$, provided condition (16) holds.

The claim follows, when applying again Proposition 2. $\square$

Although the second inequality in condition (14) is linear with respect to $h$, we will show that Example 2 cannot be directly extended in a vector way, even in the case of monoms, which justifies considering the vector case separately.

Let us consider the monome

$$h(x) := c \prod_{j=1}^n x_j^{\alpha_j},$$

where $c > 0$, and each $\alpha_j$ is even. For any positive $D$, take $q_i = \frac{D}{\sqrt{n}}$, for all $i = 1, \ldots, n - 1$, and $q_n = \epsilon \frac{D}{\sqrt{n}}$, with $\epsilon \in (0, 1)$. Then, $q = (q_1, \ldots, q_n) \in B_D$. Now, let us compute $L_q(x) \cdot x$, for $x = (1, \ldots, 1, -\theta)$. It is a quadratic polynomial with respect to $\theta$, whose discriminant $\Delta$ takes in terms of $\epsilon$ the form $\Delta = \frac{a}{\epsilon^2} + b + c\epsilon^2$. Thus, for a sufficiently small $\epsilon$, the discriminant $\Delta$ is positive, demonstrating that $L_q(x) \cdot x$ can admit negative values.

On the other hand, in the following illustrative example, we will be able to obtain a suitable local condition for $h(0) > 0$, even with an explicit $D$.

**Example 3** (condition (14)). *Let us consider*

$$h(x) := C + \prod_{j=1}^{n} x_j^{\alpha_j},$$

*where $C > 0$, and each $\alpha_j$ is even. Observe that $h$ is everywhere positive and such that*

$$L_q(x) \cdot x \geq [-\|\nabla h(q)\| \cdot \|q\| + h(q)]\|x\|^2.$$

*Hence, in order to satisfy condition (14), it is enough to obtain the inequality*

$$\|\nabla h(q)\| \cdot \|q\| \leq h(q),$$

*and since $h(q) \geq C$, it is still enough to have*

$$\|\nabla h(q)\| \cdot \|q\| \leq C.$$

*A basic majorization of the $i$-th component of $\nabla h(q)$, under the constraint $\|q\| \leq D$, is $\alpha_i D^{\Sigma_j \alpha_j - 1}$. From this, for $\|q\| \leq D$, we obtain*

$$\|\nabla h(q)\|.\|q\| \leq D^{\Sigma_j \alpha_j} \sqrt{\sum_j \alpha_j^2}.$$

*Thus, it is sufficient to choose*

$$D = \left(\frac{C^2}{\sum_j \alpha_j^2}\right)^{\frac{1}{2\Sigma_j \alpha_j}}.$$

*We have not obviously made an optimal majorization of $\|\nabla h(q)\|$ in order to obtain a simple and transparent condition. In other words, our estimation can be certainly improved for obtaining a larger $D$.*

*Let us deduce a slightly more effective condition for $n = 2$ (again not an optimal one). In this case,*

$$\|\nabla h(q)\| = |q_1|^{\alpha_1 - 1}|q_2|^{\alpha_2 - 1}\sqrt{\alpha_1 q_2^2 + \alpha_2^2 q_1^2} \leq |q_1|^{\alpha_1 - 1}|q_2|^{\alpha_2 - 1}\max\{\alpha_1, \alpha_2\}\|q\| \leq c(\alpha)D^{\alpha_1 + \alpha_2 - 1},$$

*where*

$$c(\alpha) := \frac{(\alpha_1 - 1)^{\frac{\alpha_1 - 1}{2}}(\alpha_2 - 1)^{\frac{\alpha_2 - 1}{2}}}{(\alpha_1 + \alpha_2 - 2)^{\frac{\alpha_1 + \alpha_2 - 2}{2}}}\max\{\alpha_1, \alpha_2\}.$$

*Thus,*

$$\|\nabla h(q)\| \cdot \|q\| \leq c(\alpha)D^{\alpha_1 + \alpha_2},$$

*and we can choose $D = \left(\frac{C}{c(\alpha)}\right)^{\frac{1}{\alpha_1 + \alpha_2}}$.*

**Example 4** (condition (15)). *Let us turn to the ball $B_D$ with a fixed $D$. We assume a priori that $h(q) \geq 0$ holds in a neighbourhood of 0. For instance, let $h(0) > 0$. Furthermore, suppose that, for some $c > 0$ and $p > 0$, we have*

$$\|\nabla h(q)\| \leq c\|q\|^p.$$

*Then, $|h(q)| \leq h(0) + \frac{c}{p+1}\|q\|^{p+1}$, and subsequently*

$$\|L_q(x)\| \leq \left[h(0) + c\frac{p+2}{p+1}\|q\|^{p+1}\right]\|x\|.$$

*Thus, let us still suppose that $h(0) > 1$ and $\beta \in (1, h(0))$. For (an explicit value of) $D$, we get for any $q \in B_D$ that*

$$\|L_q(x) + \lambda x\| \leq \left[\lambda - \left(h(0) + c\frac{p+2}{p+1}\|q\|^{p+1}\right)\right]\|x\|.$$

*Assume finally that $\lambda - h(0) > 1$, and take any $\beta \in (1, \lambda - h(0))$. We have*

$$\left(\lambda - \left(h(0) + c\frac{p+2}{p+1}\|q\|^{p+1}\right) > \beta\right) \Leftrightarrow \left(\|q\| \leq \left(\frac{p+1}{p+2}\frac{\lambda - h(0) - \beta}{c}\right)^{\frac{1}{p+1}}\right).$$

*After all, we can take*

$$D \leq \left(\frac{p+1}{p+2}\frac{\lambda - h(0) - \beta}{c}\right)^{\frac{1}{p+1}}$$

*in order to satisfy condition (15). By the optimization with respect to $\beta$, we can readily check that any $D$ satisfying*

$$D < \left(\frac{p+1}{p+2}\frac{\lambda - h(0) - 1}{c}\right)^{\frac{1}{p+1}}$$

*can be chosen for it. The last step is to specify $D$ such that $h(q) \geq 0$, for every $q \in B_D$.*

## 5. Conclusions

Under the assumptions of Theorems 1 and 2, the obtained limit-periodic solutions are obviously also almost-periodic. On the other hand, if the forcing terms $\{r_t\}$ in Equation (6) are almost-periodic (or, in particular, quasi-periodic), then one should proceed in a different manner in order to get an almost-periodic (resp. quasi-periodic) solution. However, if the forcing terms $\{r_t\}$ in Equation (6) are at the same time limit-periodic and quasi-periodic, then they become simply periodic (see [4], Theorem 2 and [5], Remark 4). In this very special case, the existence criteria for periodic solutions can be significantly improved. Concretely, conditions (9), (14), and (15) can be reduced into $h(x) \geq 0$, for $x \in B_D$.

Observe that, in the special case of the limit-periodically forced logistic equation (i.e., $h(x) := -\mu x + \mu - \lambda$, $\mu > 0$), namely

$$x_{t+1} + \mu(x_t - 1)x_t = r_t,$$

condition (9) takes the form

$$\exists D > 0 \text{ such that, for some } \lambda \in (1, \mu),$$
$$\forall x \in [-D, D]: \ -2\mu x + \mu - \lambda \geq 0.$$

Condition (10) is the same as above, i.e.,

$$\|\underline{r}\|_\infty \leq \frac{\lambda - 1}{\lambda + 1}D.$$

One can readily check that they can be satisfied for $D = \frac{\mu - \lambda}{2\mu}$, $\mu > \lambda > 1$, and

$$\|\underline{r}\|_\infty \leq \frac{\lambda - 1}{\lambda + 1} \cdot \frac{\mu - \lambda}{2\mu}.$$

For example, taking $\mu = 3.5$ and $\lambda = 2$, we have $D = \frac{3}{14}$, and subsequently $\|\underline{r}\|_\infty \leq \frac{1}{14}$.

Let us finally note that if condition (9) holds on the whole line, like e.g., for $h(x) := \frac{\pi}{2} + \arctan x$, then Equation (6) admits a limit-periodic solution for any limit-periodic forcing $\{r_t\}$. In the special case of $h(x) := \frac{\pi}{2} + \arctan x$, the second inequality in condition (9) namely holds because

$$[h'(x)x + h(x)]' = \frac{2}{x^4 + 2x^2 + 1} > 0,$$

by which $h'(x)x + h(x)$ is strictly increasing, jointly with

$$\lim_{x \to -\infty}[h'(x)x + h(x)] = \lim_{x \to -\infty}\left[\frac{x}{1 + x^2} + \frac{\pi}{2} + \arctan x\right] = 0.$$

More generally, a sufficient condition for satisfying condition (9), for all $x \in \mathbb{R}$, takes the form:

$$h(x) \geq 0, \ [h(x)x]'' \geq 0 \ \text{ and } \ \lim_{x \to -\infty}[h(x)x]' \geq 0.$$

In the vector case, the same is true, provided condition (14) or condition (15) holds, for all $q \in \mathbb{R}^n$.

**Author Contributions:** The contributions of both authors (J.A. and D.P.) are equal. All the main results and illustrative examples were developed together.

**Funding:** This research was funded by Grant Agency of Palacký University in Olomouc grant number IGA_PrF_2018_024 "Mathematical Models".

**Conflicts of Interest:** The authors declare no conflict of interest.

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
