# Peer review of "Note on Limit-Periodic Solutions of the Difference Equation xt + 1 − [h(xt) + λ]xt = rt, λ > 1"

_axioms, doi:10.3390/axioms8010019_

Round 1

Reviewer 1 Report

The authors mainly study uniformly limit periodic solutions of the treated difference equations in the scalar and vector case (see Eq. (6) in the manuscript). They identify conditions which guarantee that the considered equations have uniformly limit periodic solutions satisfying certain inequalities (see Theorem 1 and Theorem 2 in the manuscript). Several illustrative examples are mentioned as well.

I have found the results new and interesting. The presentation of the results is clear and the used arguments are correct and easy to understand. The manuscript contains almost no misprints or other formal flaws. I think that it could be published in the present form. Thus, I strongly recommend the manuscript for publication.

I have only two unimportant suggestions which follows:
- to replace "our recent paper" by at least the full title of the paper in Abstract;
- to omit "$(- \mu x + \mu - \lambda \ge 0 \Leftarrow),$" on p. 8_6, i.e., $h(x) \ge 0$ (it is a bit redundant).
Of course, the manuscript can be published in the present form.

Author Response

I accepted both suggestions and incorporated them into the revised version.

Reviewer 2 Report

In the paper, the authors  developed  their previous results, they established some sufficient conditions  for  the existence of limit-periodic solutions of the difference equation from the title and both  scalar and vector cases are included in their discussion. Finally, Several simple illustrative examples are provided. However, I suggest some key related literatures to be added to the paper:

[1] Almost periodic solution for a new type of neutral impulsive stochastic Lasota–Wazewska timescale model, Applied Mathematics Letters, Volume 70, August 2017, Pages 58-65.

[2] Almost periodic dynamics for impulsive delay neural networks of a general type on almost periodic time scales, Communications in Nonlinear Science and Numerical Simulation, Volume 36, July 2016, Pages 238-251.

[3] Almost periodic solutions of differential equations with piecewise constant argument of generalized type, Nonlinear Analysis: Hybrid Systems, Volume 2, Issue 2, June 2008, Pages 456-467

[4] Periodic motions generated from non-autonomous grazing dynamics, Communications in Nonlinear Science and Numerical Simulation, Volume 49, August 2017, Pages 48-62

After this minor revision, I will recommend the paper for publication.

Author Response

I accepted the suggestion and incorporated all the suggested references into the revised version.
